# OpenReview forum: "Beyond Natural Language: Invented Communication in Vision-Language Models"
_ICLR.cc/2026/Conference — ICLR 2026 Conference Withdrawn Submission_

### Official Review · Reviewer_4cAN · 2025-10-29

**Soundness:** 2
**Presentation:** 2
**Contribution:** 2
**Rating:** 2
**Confidence:** 4

**Summary:**

This paper explores whether vision–language models (VLMs) can come up with their own communication codes that are shorter and harder to interpret than natural language. The authors set up a referential game where agents must identify flags and compare three communication conditions: NATURAL, EFFICIENT, and COVERT. Without any finetuning, just prompting, they show that models can invent novel tokens, sometimes coordinate with each other, and achieve high accuracy per character. They also add analysis on word novelty, corpus similarity, and run a small human study.

The idea is timely and fun, asking if models will “cheat” natural language when pushed to be efficient or private. That said, I think the current results mainly show that models can build ad-hoc codes for specific games, rather than fully demonstrating new languages. The efficiency claims are overstated, the covertness experiments are weak, and the heavy reliance on “informed sender” setups confounds the conclusions. Still, this is an interesting seed of a paper if reframed more modestly.

**Strengths:**

- The paper asks a simple but compelling question: can VLMs invent communication schemes under pressure?
- The “prompt-only” design is a strength; it avoids heavy finetuning and shows what large pretrained models can do off the shelf.
- Using multiple models (GPT-4o, Qwen, Pixtral).
- The linguistic analyses (novel word rate, similarity metrics, UMAP) show some effort to go beyond raw accuracy.
- The connection to safety (covert communication, detectability) is relevant and potentially impactful.

**Weaknesses:**

- **Efficiency metric is shaky.**
  The core claim that invented codes “surpass natural language in efficiency” rests on the metric of accuracy per character. This is not a standard or theoretically grounded measure. Characters are a poor proxy for information, and combining them with accuracy creates odd incentives, e.g., an agent can trivially assign one unique character per candidate in a single round and score very high, even though this “code” has no reusable semantics. Without bits-per-message, compression ratios, or token-level measures, it’s not clear the paper is actually measuring efficiency in any meaningful linguistic sense.

- **Design confounds (informed sender).**
  Most results use the “informed sender” setting, where the sender sees not only the target but also the distractors. This lets the sender design one-off codes optimized for that specific round, rather than general-purpose language. That setup inherently advantages ad-hoc symbol assignment and explains much of the efficiency gains. Without this setup, the evidence for emergent codes is weaker, and the comparison to natural language is unfair.

- **Overstated claims of coordination.**
  The claim that agents “spontaneously coordinate” without sharing a codebook is overstated. The examples suggest they converge on very similar invented tokens, but this is plausibly due to shared pretraining biases, prompt wording, or simple lexical priors rather than genuine emergent alignment. To really support this, the authors would need controls such as forcing disjoint vocabularies, running multiple seeds, or showing transfer across independently trained agents.

- **Covertness not rigorously tested.**
  The paper’s “covert” evaluation relies on an overseer agent that is asked to interpret the invented codes as if they were natural language. Unsurprisingly, this fails. But a true test of covertness should involve an adversarial listener who can adapt, e.g. given a few labeled examples or allowed to train a small classifier. Without this, the evidence for covertness is weak, it currently shows only that the codes are not trivially human-readable, which is much less meaningful.

- **Generalization missing.**
  The invented codes are tested only on the specific world they were generated in. There is no evidence that they transfer to new distractor sets, new compositions, or permuted configurations. This limits the claim that these are “languages” at all; they may simply be local encodings with no reusable semantics. A compositional generalization test would have been essential here.

- **Statistical rigor is low.**
  Most tables report point estimates with no confidence intervals, no tests of significance, and no discussion of variance across seeds. Results may be brittle to decoding settings, randomness, or prompt formulation. The lack of robustness checks makes it hard to trust the reported differences.

- **Baselines are missing.**
  The study does not compare against simple human-designed shorthand baselines, like abbreviations (“blu tri L-diag”) or handcrafted codebooks. These would probably achieve similar or higher efficiency while remaining interpretable. Without these, it’s hard to know if the VLMs are really doing anything surprising.

- **Reproducibility concerns.**
  Many of the most striking results involve GPT-4o, a closed model whose behavior changes over time (and outdated as of today). The authors don’t provide seeds, API version details, or decoding parameters. This makes it difficult to reproduce or verify the findings.

**Questions:**

- **On efficiency metrics:** Could you provide results in terms of tokens or bits per message rather than characters? For example, rate–accuracy curves using a fixed tokenizer (like GPT-2 BPE) would make the efficiency claims much stronger. How would your conclusions change if measured this way?

- **On the informed-sender setup:** How much of the efficiency advantage disappears without the informed-sender? Can you show side-by-side results (informed vs not) to clarify whether the emergent codes are general-purpose or simply exploiting distractor knowledge?

- **On coordination claims:** In your “spontaneous coordination” experiments, how robust is the effect across seeds, prompts, or when sender and receiver are forced to use disjoint alphabets? Could you quantify lexical overlap across runs to rule out prompt or prior biases?

- **On generalization:** Do the invented codes transfer to unseen candidate sets, new synthetic worlds, or permuted feature configurations? If not, should we think of them as “ad-hoc codes” rather than “languages”? A held-out generalization test would clarify this.

- **On covertness:** Right now, covertness is defined as “hard for a naive overseer.” Could you add a stronger baseline, e.g. an adversarial listener trained on a handful of (message, image) pairs, or a meta-receiver using embeddings to decode? What is the sample complexity needed to crack the code?

- **On human study:** Please provide more details: how many trials per participant, were there attention checks, and how were participants compensated? Also, were humans allowed to see multiple rounds and build up a mapping, or was each trial independent?

- **On baselines:** Did you try simple natural-language compression (abbreviations like “blu tri L-diag”) or a handcrafted symbolic codebook (e.g. color initials + shape initials)? How would these compare in your accuracy-per-character metric?

- **On robustness:** Could you report variance across random seeds, decoding temperatures, and prompt templates? This would help assess how stable the results are, especially with nondeterministic models like GPT-4o.

-  **On real flags vs synthetic:** You note that using real flags leaks pretrained knowledge. Did you completely separate results from real vs synthetic worlds, or do any of the headline claims rely on real-flag cases?

- **On open models:** Since GPT-4o is closed and drifts over time, could you highlight whether the main findings (efficiency, covert codes) still hold on Qwen2-VL or Pixtral alone? This would make the claims more reproducible for the community.

---

### Official Review · Reviewer_9Tyv · 2025-10-30

**Soundness:** 2
**Presentation:** 3
**Contribution:** 2
**Rating:** 4
**Confidence:** 3

**Summary:**

The paper concerns the capabilities of vision language models (VLMs) in inventing task-specific languages. The specific setting being investigated is a cooperative game: A sender has to describe an image in such a way that a receiver can correctly select it from a set of candidate images (the sender has access to the same set of images). Via appropriate prompting, various settings are tested: (1) the sender describes the image in natural language, with different soft constraints on the number of words it is allowed to use (2) the sender is instructed to invent a new language, which is shared—in addition to the message describing the image—with the receiver (3) similar to before, but only the message, not the language, is shared with the receiver (4) the models are additionally instructed to make the language covert, i.e. not easily interpretable by outside or even human agents.

**Strengths:**

- **Thorough Experimentation:** The authors test their hypotheses across three different VLMs (Gpt, Qwen, Pixtral) , two dataset variants (real and synthetic flags), and multiple protocol-sharing conditions (local vs. shared).
- **Human Evaluation:** The inclusion of a human evaluation study (Section 5.4) provides an important baseline for interpretability and supports the paper's insight, namely that models can invent languages more interpretable to themselves than to humans.
- **Rigorous Analysis:** The study goes beyond simple task accuracy. It includes metrics like "accuracy per character" (which accounts for models "cheating" length constraints) and a detailed "New Word Rate" analysis.
- **High-Quality Appendices:** The paper is supplemented by appendices that include full experimental prompts (Appendix E) , dataset details (Appendix F) , and extensive language similarity analysis (Appendix I).

**Weaknesses:**

The paper frames its investigation with grand concepts like the linguistic relativity hypothesis and quotes Wittgenstein on language limiting one's world. However, the "invented languages" are operationally simple {concept: description} lexicons, explicitly prompted for in the "efficient" condition (Appendix E.2). I would not call that the emergence of a new language with novel symbolic or computational properties, as suggested by the introductory quote from Silver & Sutton. The study does not investigate the emergence of syntax, compositionality (beyond simple token-mapping), or other fundamental properties of language. In addition, the paper's use of "emergent communication" and "spontaneous invention" is questionable. While a genuinely spontaneous invention of new words (e.g., 'blorple') is noted in the "natural" condition under tight constraints, the primary "efficient" and "covert" experiments rely on explicit, structured instructions to invent a language. This instructed setup limits the claims that can be made about _emergence_ in the classical sense.

The results from the "locally-invented-language" setting are difficult to interpret. First, the finding that identical models (e.g., QWEN-QWEN) can coordinate without sharing a protocol is not surprising. As Appendix A.4 clearly shows, two QWEN models given the same prompt and images produce identical languages. It is expected identical models with the same prompt generate similar outputs, if deterministic sampling (e.g., temp=0) is used the output is even guaranteed to be identical. The authors should clarify the sampling parameters used, as this result may be an artifact of the generation process rather than a profound finding about coordination. I think the results of the 'overseer' model in these settings is more meaningful, but also less conclusive.
Second, it is unclear if the Sender agent is aware of the sharing condition (local vs. shared) when it invents its language. The prompts in Appendix E do not seem to specify this. An agent's strategy for language invention could be fundamentally different if it knows its protocol must be independently re-created by a partner, versus knowing it can simply share the protocol. This ambiguity in the experimental design seem to me like a methodological concern.

These considerations lead me to vote for a weak reject at the moment.

**Questions:**

- Could the description length also be measured in bits (cumulative NLL of the tokens, computed either by the sender or the receiver), instead of word or character count?

---

### Official Review · Reviewer_BPTv · 2025-11-01

**Soundness:** 2
**Presentation:** 3
**Contribution:** 2
**Rating:** 4
**Confidence:** 3

**Summary:**

This paper investigates whether VLMs can develop communication protocols that outperform natural language using a referential game on flag images. The authors show that under constraints models produce novel tokens and code-like messages, with same-architecture pairs communicating best; however, the central claims would benefit from more rigorous metrics, fairer comparisons, and a more mechanistic account.

**Strengths:**

I like the core idea. Probing whether VLMs invent compact, “private” codes under efficiency/covert pressures is both original and timely for safety/alignment, where covert channels are a real concern.

The experimental design is simple and replicable. A controlled referential game cleanly isolates effects and makes comparisons straightforward. The results of same-architecture vs. cross-model pairing are interesting.

**Weaknesses:**

1. Same-architecture advantage is not causally isolated. The finding that same-architecture pairs interpret each other’s messages better—suggesting a shared internal representation—may stem from shared implementation choices (tokenizer, vision backbone, prompt template, decoding temperature) rather than deeper representational overlap. Please add cross-swap ablations (e.g., swap tokenizers, mismatch/replace the vision trunk, freeze different submodules, vary decoding strategy) or at least discuss these confounds more explicitly.

2. Informed sender assumption is overly simplified. The Informed sender sees all 10 candidates, which is uncommon in real communication (the sender typically cannot preview the receiver’s distractors). This lets the sender anticipate exactly where ambiguity arises and likely inflates the gains of ultra-compact codes; in practice—when the sender does not know the candidate set—the benefit may be much smaller.

3. Accuracy–efficiency trade-off needs clearer framing. The abstract claims efficient languages are shorter descriptions yet covert enough to remain unintelligible—implying a win-win. In reality, Table 2 shows accuracy drops from 97% to 71% (-26%) for 4.5× compression. Please clarify the utility model that justifies this loss and when such a trade-off is acceptable for target applications.

4. AI vs. human comparison is not very informative. Humans are evaluated with no feedback or learning—why not offer a brief “decoding” phase? Please report accuracy after a short tutorial or a small self-study set. Also, in the current 10-way setting, chance is 10%, yet humans reach ~43%; what does this imply—are participants exploiting visual regularities or residual natural-language cues?

**Questions:**

1. The paper's central claim—that models "invent language"—lacks rigorous support. Is this truly language invention or just embedding projection? Do the generated codes satisfy some basic linguistic properties? Without a more careful discussion, framing it as “invented language” risks overstating what’s shown.
2. I’m curious: if you make small changes to the sender’s code, does the receiver still decode reliably? For example, lightly perturb the token sequence (swap/insert/delete) or inject minor noise into the symbolic code—does accuracy hold up or collapse?

---

### Official Review · Reviewer_po6M · 2025-11-02

**Soundness:** 1
**Presentation:** 1
**Contribution:** 2
**Rating:** 2
**Confidence:** 5

**Summary:**

The paper explores referential games using pre-trained models and the zero-shot prompt setting. There are different prompts associated with different agent roles, and the inputs are flags (real or synthetic). Experiments are conducted in different settings (natural, efficient, covert) and with three different models (gpt, qwen, pix). Results show expected trends across models and setting, indicating this could be a useful setting for explore emergent communication protocols.

The main weaknesses in the paper are (1) that it contains very strong and overly broad claims given the presented information. For example,
- "this work is the first to demonstrate the ability of vision-language models (VLMs) to invent novel languages and use them effectively to describe visual inputs"
- "models can invent languages that are more interpretable to themselves than to humans"

And (2) that its fairly limited in results and analysis compared to previous work in emergent communication.

**Strengths:**

S1. The paper explores whether pretrained models can be used to study emergent communication even in the zero-shot prompt setting. The results are encouraging along these lines. This could serve as a template for future work, e.g. exploring few-shot or different input types.

S2. The paper explores multiple models and settings. The details for such settings are well documented in the appendix, including with output examples.

S3. I believe this is the first paper that uses flags to study emergent communication. There are some conveniences to this data. Although it was not directly compared against other existing dataset options.

**Weaknesses:**

W1. The paper requires some major clarifications for this statement: "this work is the first to demonstrate the ability of vision-language models (VLMs) to invent novel languages and use them effectively to describe visual inputs"

Many (even most) of the previous work in emergent communication was with language models that integrate a visual component. Also, previous work often relies on continuous high bandwidth messages, binary messages, or even images [3], which arguably fit the language descriptions used in this paper. There has also been studies focused on language complexity beyond novel words [4]. The clarification that is needed here:
- How is the VLM described here different from previous work?
- What is the difference between a novel language here compared previous work?

Essentially, this statement is far too broad given there is so much previous work that has focused on the multi-modal setting and on inventing novel languages.

[1] Learning Multiagent Communication with Backpropagation https://proceedings.neurips.cc/paper_files/paper/2016/file/55b1927fdafef39c48e5b73b5d61ea60-Paper.pdf
[2] Emergent Communication in a Multi-Modal, Multi-Step Referential Game https://arxiv.org/abs/1705.10369
[3] Learning to Play Guess Who? and Inventing a Grounded Language as a Consequence https://arxiv.org/abs/1611.03218
[4] Emergent Linguistic Phenomena in Multi-Agent Communication Games https://arxiv.org/abs/1901.08706


W2a. I think there is some ambiguity in this paper about whether the model can effectively guess the meaning of words in a compressed / derived new language, or if they are introducing a new technique that does the language invention. For example, I took the standalone text "mul kiv cir fep qua mak" from Figure 1 and ask GPT-5 to guess its meaning. The response from gpt-5:

My request:

    mul kiv cir fep qua mak

    Can you guess this meaning? An AI produced this when asked to invent a language for a referential game.

The response from gpt-5:

    Fun! Total guess—but in many referential games each CVC token maps to a feature (det, color, shape, size, relation, landmark). Reading yours that way:

    mul kiv cir fep qua mak ≈
    “the small green circle left of the square”

    - mul → determiner (“the/this”)
    - kiv → color (green)
    - cir → shape (circle)
    - fep → size (small)
    - mak → spatial relation (left-of)
    - qua → landmark object (square)

    I’m not certain (these languages are arbitrary), but “cir” strongly hints at circle, and having both cir and qua suggests “target + landmark,” with mak as the relation. If you share the inventory of shapes/colors/relations from the game, I can pin it down exactly.

There is quite a lot of overlap with gpt-5's guess here and the language description in Figure 1. Usually, the protocol in emergent communication forces two agents to collaborate and invent a language together, but here it seems like the agents can simply guess the meaning of words independent of the protocol.


W2b. This statement also seems needing of clarification "models can invent languages that are more interpretable to themselves than to humans"

- For one, based on prior work, the default is that models develop a language that they can understand even if a human can not. It is usually more work to make the language interpretable to humans.
- Using the example from W2a, it does seem parts of the language are guessable. I posit that humans who have a specialized knowledge of building languages like this could effectively guess the meaning of words in the language.


W3. The game is limited to the one turn setting (many previous work explore the conversational setting), and the language analysis in sec 5.5 does not go very deep. The paper's definition for new words is very permissive (table 3 even contains some new words such as "rcvr", "desc", "ovrsr"). Also, there is no notion of "control group" for Figure 4, and it could be that many instances of three prompts are roughly clustered in a similar way. In previous work, analysis often goes further, e.g. comparing the distribution of words to zipf's law, tying message complexity to input difficulty, and so on.

**Questions:**

n/a

---

### Note · Authors · 2025-11-20

**Comment:**

### Withdrawal of ICLR Submission

Dear Reviewers,

Based on your insightful comments, suggestions, and overall assessment, we have decided to withdraw our paper at this stage. We plan to continue improving the work and hope to submit a stronger version in the future.

We sincerely thank you for the time and effort you dedicated to reviewing our paper. Your feedback is greatly appreciated.

Best regards,
The Authors

**Withdrawal Confirmation:**

I have read and agree with the venue's withdrawal policy on behalf of myself and my co-authors.